# The Potential Role of Stakeholders in the Energy Efficiency of Higher Education Institutions

**Rubén Garrido-Yserte ***(ID) **and María-Teresa Gallo-Rivera ***(ID)

Department of Economics and Business Management and Institute of Economy and Social Analysis,
University of Alcalá, Victoria Square, 2, 28802 Alcalá de Henares, Madrid, Spain
* Correspondence: ruben.garrido@uah.es (R.G.-Y.); maria.gallo@uah.es (M.-T.G.-R.);
  Tel.: +34-918855193 (R.G.-Y.); +34-918855112 (M.-T.G.-R.)

**Abstract:** Higher education institutions (HEIs) have a huge potential to save energy as they are significantly more energy-intensive in comparison with commercial offices and manufacturing premises. This paper provides an overview of the chief actions of sustainability and energy efficiency addressed by the University of Alcalá (Madrid, Spain). The policies implemented have shifted the University of Alcalá (UAH) to become the top-ranking university in Spain and one of the leading universities internationally on environmentally sustainable practices. The paper highlights two key elements. First, the actions adopted by the managerial teams, and second, the potential of public–private collaboration when considering different stakeholders. A descriptive study is developed through document analysis. The results show that energy consumption per user and energy consumption per area first fall and are then maintained, thereby contributing to meeting the objectives of the Spanish Government's Action Plan for Energy Saving and Efficiency (2011–2020). Because of the research approach, the results cannot be generalized. However, the paper fulfils an identified need to study the impact of HEIs and their stakeholders on sustainable development through initiatives in saving energy on their campuses and highlights the role of HEIs as test laboratories for the introduction of innovations in this field (monitoring, sensing, and reporting, among others).

**Keywords:** energy consumption; higher education institution; energy efficiency indicators; green campus; social responsibility; Spain

---

## 1. Introduction

Currently, higher education institutions (HEIs) worldwide have incorporated sustainability development strategies into their study programs, research, operations, dissemination, assessment, and reporting. Undoubtedly, the international implementation scheme of the United Nations (UN) a Decade of Education for Sustainable Development (2015–2014) has boosted the integration of the principles of sustainable development (SD) in all aspects of HEIs [1]. The institutionalization and incorporation of these strategies has been progressive and has had to overcome the initial resistance of the main stakeholders [2,3]. HEIs are significantly much more energy-intensive in comparison with commercial offices and manufacturing premises, so improving their energy efficiency becomes a fast and cost-effective method of achieving targets in the reduction of gas emissions and can foster their economic growth. The main advantages of these strategies are cost reductions, environmental protection, better public health, and economic sustainability [4,5].

The energy efficiency and carbon reduction initiatives of HEIs are conditioned by both national programs and the main trends in HEIs (increases of students, associated complexity of research and

teaching activities, and the intensive use of equipment, resulting in an increase of energy demand and consumption on campus) [6].

The main influences on the policies and initiatives of HEIs concerning energy efficiency and carbon reductions come from both external and internal sources. The former includes European Union (EU) directives, such as the EU Emissions Trading Scheme (EU ETS), the Environmental Performance of Buildings Directive (EPBD) (2010/31/EU) on energy efficiency in buildings, and the 2012/27/EU energy efficiency directive. The national regulatory framework in the case of Spain includes the Action Plan For Energy Saving and Efficiency 2011–2020, the application of a Technical Building Code (RD 314/2006), the Basic Procedure for Energy Performance Certification of New Buildings (RD 47/2007), the Regulations on Heating Installations in Buildings (RD 1027/2007), and the Building Certification Regulations (RD 235/2013) which replaced the RD 47/2007 [7].

The latter sources include increased energy costs, initiatives in the arena of corporate social responsibility, attention to legal obligations, maintenance of economic competitiveness, concerns for the environment, the capacity to access financing sources, and the interest for corporate image [6]. Both kinds of factors determine the presence of an "energy efficiency gap" in the organizations, referring to a potential improvement of energy efficiency or, in other terms, the difference between a cost-minimizing level of energy efficiency and the level of energy efficiency actually implemented. This gap is also conditioned by economic factors and, especially, by organizational and institutional factors. The main organizational barriers for implementation of energy efficiency interventions in HEIs include access to financing sources, bounded rationality, hidden costs, imperfect information, risk and uncertainty, and non-aligned incentives [4].

Furthermore, the insights and perceptions of staff and students are crucial to the success of energy efficiency interventions. Generating a greater responsibility of the occupants towards energy conservation requires three main aspects to be taken care of: transmitting a clear conservation message, improving their participation in energy reduction schemes, and providing correct information about the use of energy [8].

The University of Alcalá (UAH), considered to be one of Europe's oldest universities, is aware of the environmental impact of its activities and, especially, of its energy consumption. The UAH is committed to sustainable development and is actively involved in finding solutions to environmental problems. It promotes the efficient use of energy through programs of energy conservation, boosting use of renewable energy and encouraging actions to raise awareness and involvement among the entire academic community. In the last few years, the UAH has been involved in several energy-saving measures to minimize its impact, which is of special relevance as stated in its 2003 Environmental Policy Statement.

In the last few years, a group of different external studies and audits have been carried out by experts to grasp more fully the University's energy needs and to set ambitious goals for cutting down consumption and replacing current energy sources with those that are cleaner and more sustainable. These policies have allowed the UAH to become, according to the Green Metrics World University Ranking, which assesses the sustainability policies of universities worldwide, the top ranking university in Spain and one of the leading universities internationally for environmentally sustainable practices.

However, in recent years, it seems that, despite good results, progress has stagnated. The stabilization in the number of students and the implementation of inefficient $24 \times 7$ services in heritage buildings have led people to demand new measures and investments in the future. A strong management commitment is needed to find the most innovative solutions in the field of energy efficiency. Innovative public procurement would seem to be an option that could be re-explored in an area where the most innovative solutions are in the private sector.

This paper provides an overview of the UAH's chief actions in sustainability matters and energy efficiency to date and introduces the main future action lines.

Two research questions were proposed to analyze the potential of implementing energy saving initiatives in HEIs, taking into account the perspective of different actors:

RQ1: What kind of actions can be implemented on university campuses in order to achieve effective energy saving along with a substantial improvement of the perceptions and the impacts of the different university community actors? What is the role of public–private partnerships in the context of financial or budgetary constraints?

RQ2: Which are the main indicators of energy saving that could be considered in an impact evaluation of HEIs on sustainable development?

Nowadays the relevance of the energy efficiency is undoubtable. The importance of the energy efficiency in HEIs is key to optimize the resources for the benefit of their academic community as well as the environment. The relevance of HEIs is threefold. First, it raises awareness about the importance of energy efficiency activities and the implementation of energy saving measures. Secondly, HEIs are places of high electricity and gas consumption due to their multiple installations, for many hours a day, many days a year, and with a large number of people. Thirdly, HEIs could be good laboratories for the introduction of industry innovations. They have an adequate dimension with activities similar to a small city; having all the elements of an urban concentration (roads, street lighting, service and transport vehicles, energy-intensive facilities, etc.) and its inhabitants (students, teachers and staff) are, a priori, more favorable to innovation and monitoring can be more effective.

This paper has a novelty that focuses on a key sector, HEIs, and contributes to the comprehension on how universities may implement effective measures of energy efficiency involving their main stakeholders, by analyzing the UAH's energy efficiency program results and its national and international recognized good practices, referent in energy saving in HEIs campuses.

Section 2 offers a review of the specialized literature in energy efficiency and carbon reduction programs, especially in the context of HEIs, in addition to the key HEI trends that influence the patterns of energy consumption and saving. The data and the methodology used as well as the case description are presented in Section 3. Section 4 analyzes the results obtained regarding UAH's interventions in respect of energy efficiency and presents an approximation of UAH's contribution to climate change. The discussion of the UAH's future challenges to improve energy efficiency is presented in Section 5. Finally, Section 6 presents the concluding remarks.

## 2. Background and Literature Review

### 2.1. Current Situation of Energy Efficiency and Carbon Reduction in Spain

The term energy efficiency is much used in public policy and refers to a set of "policies that seek to reach a balance in sustainable development, competitiveness, and secure supply, mainly by promoting energy efficiency and the use of renewable energies and other directives and documents directed at the energy sector" [7].

Over the years, the European Commission has implemented a set of directives and regulations to strengthen the EU energy policy. What have become known as 20/20/20 targets are part of the current EU strategic energy policy goals, which include increasing energy efficiency to achieve 20% savings in EU energy consumption by 2020.

Quantitative indicators are used often to assess the progress toward energy efficiency targets. The energy intensity ratio (consumption of energy related to its gross domestic product) shows the extent to which energy consumption has become more sustainable, illustrating the reduction in energy used to generate one unit of activity in the case of GDP remaining constant. Other indexes are also used to provide a more realistic evaluation of improvements in energy efficiency [9].

The different EU Member States have made different levels of progress toward these goals. Overall, the trend is positive, but their efforts are not enough to achieve the EU goal of 20% energy saving. The Spanish progress in energy efficiency issues, and particularly in the building sector, is slower than in other EU Member States and has been intensified by the economic crisis, the demotivation of its citizens, the decentralization of power in energy matters, and the huge stock of houses for sale. In the household sector, Spain achieved an energy efficiency gain of 12.6% in 2000–2010, which is below the

EU-27 average of 15.3%; moreover, evidence has shown that Spanish housing markets capitalize the value of energy efficiency [7,10].

The last report of Regulatory Indicators for Sustainable Energy (RISE 2018) published [11], indicates that Spain is located in the 22nd position in the energy efficiency ranking, well behind some countries of similar socioeconomic level. This reveals that, in practice, public policies relating to the efficient consumption of energy are not yet a priority in Spain's energy policies.

### 2.2. The Role of HEIs for Sustainable Development and Key Trends in the HEIs' Sector

Universities can be regarded as "small cities", and their activities can have great impacts in terms of environmental pollution and degradation. Universities also have special societal responsibility with regard to knowledge, skills, values, and social awareness of sustainability [12,13].

The contributions of HEIs to the Sustainable Development Goals (SDG) stem from their five functions as educators—providing knowledge; trainers—providing professional training to manage the needs of government and other stakeholders; researchers—producing data and analysis for policymakers; facilitators—contributing to regional development and international cooperation; and enablers—improving social and intellectual development and the well-being of society. Specifically, in the area of sustainability, HEIs have an impact in preparing individuals to meet the needs of modern economies across global, national, regional, and local levels; i.e., creating a competitive, adaptable, sustainable, and knowledge-based economy—particularly a green economy—with benefits in terms of social equity and well-being, while reducing their environmental impacts [14].

The initiatives to implement Sustainable Campuses or Green Campuses have grown in the last ten years across the world. However, there are different definitions of what a sustainable university campus is. Therefore, interpretations and strategies to achieve a sustainable campus vary from one university to other.

Next are presented some definitions about what a sustainable or a green campus is.

"A green campus is one which addresses environmental challenges in all its fields of activity: administration, research and education. Higher education institutions are in an outstanding position to act as incubators, role models and multipliers for sustainable development among researchers, people in leadership positions, and in wider society" [15].

" . . . a healthy campus environment, with a prosperous economy through energy and resource conservation, waste reduction, and efficient environmental management, promotes equity and social justice in its affairs and exports these values at community, national, and global levels" [12].

Considering this framework, for some universities, a sustainable campus is defined by either an environmental plan or an environmental statement. Others consider the signing of a national or international declaration aimed to tackle the sustainability challenge, and others conduct their own approaches, including green building initiatives, ISO 14001, and Environmental Management Systems (EMSs) [12,15]. In the latter, the universities that have the practice of diffusion of sustainability reports have improved their visibility, mobilizing resources to improve their endowments and raising funds for future sustainability activities. Therefore, the sustainability reporting could become a tool with huge potential to increase the engagement of the main stakeholders [16,17].

In any case, the evidence indicates that a three-pronged strategy that include university EMSs; public participation and social responsibility with the real implication of key stakeholders, and promoting sustainability in teaching and research, could be the framework to foster a more suitable approach to achieving campus sustainability in a systematic and integrated way [12,18].

It is evident that the HEIs' sustainability practices have been increasing, and they are prominent especially in Europe, the US, and Canada in addition to Australia, Asia, South America, and Africa. Some guides and good practices in HEIs are analyzed in literature and by international organizations as The Organisation for Economic Co-operation and Development (OECD) and International Alliance of Research Universities (IARU) [13,19,20]. Furthermore, well know are the energy efficiency interventions

in HEIs as Coimbra, Portugal [21,22], the United Kingdom [6,8], the United Kingdom and South Africa [23], Canada [24], and Universiti Teknologi Malaysia [25].

Although the role of HEIs in environmental, economic, and social sustainability is key, the level of engagement is different; there is a strong focus on activities related to environmental sustainability, whereas there is a weaker focus on social and economic sustainability [13]. In the environmental sustainability area, the implementation of EMSs is more frequent, and they comprise either whole campuses or part of them and those that include either all environmental issues or just some key issues, such as either water or energy. Some HEIs have connected their financial systems with financial incentives to boost sustainability behaviors; and the improvements in the teaching and research activities related to sustainability development are relevant.

In terms of economic sustainability, in the less-developed area, there are practices related to connecting the financial system with new purchasing procedures and developing financial incentives to promote sustainability behaviors. These include new forms of contracts with providers, premiums to staff who participate in the sustainability projects, contribution in innovation through building and infrastructure planning, and increasing research capacity to address the organizational challenges related to sustainability matters.

Regarding social sustainability, there are different level of social responsibility of HEIs related to staff and student well-being and the relationship with local communities; practices related to prevention and mental healthcare, volunteer activities, outreach programs, and approaches of Corporate Social Responsibility into HEIs with the participation of key stakeholders, including the local governments and interaction with small and medium size business enterprises (SMEs).

In the Spanish case, the engagement of universities in sustainability issues has received a decisive stimulus since 2002, when the Spanish University Rector's Committee (CRUE) approved the creation of a working group of Environmental Quality and Sustainable Development in a group of thirty universities (Sustainability-CRUE). In turn, in 2009, this group created a special commission focused on environmental quality, sustainable development, and risk prevention (CADEP-CRUE). Since then this commission has developed seminars to exchange experiences, encourage good practices, and develop joint projects [26,27]. Since January 2014, the Sustainability-CRUE Chair has been Professor Galván, Rector of the UAH. Furthermore, the creation of an Inter-university network of teaching and research in education for sustainability by the CADEP-CRUE represents an important point of reference about materials and teaching experiences in the field of sustainability in university teaching [28].

In the 8th Seminar of the CADEP-CRUE (2009) about [27], the Spanish universities recognized that HEIs have huge potential to saving energy "buildings are not the ones that waste energy, but people in their daily use". The availability of accurate statistical data on energy consumption of centers and buildings, would allow for developing reliable indicators before initiating any intervention, measuring the starting situation and after the intervention. It is also appropriate that each university has an energy manager to secure the commitment from institutional managers, to get better energy certifications for its buildings, and to both develop and manage their energy efficiency plans. Besides, projects must include a post-construction energy maintenance and management plan. The university contracting and bidding process must include all the guarantees necessary to make buildings sustainable. Finally, a close relationship between universities and companies is fundamental for sustainable energy and also serves as a test bed for new technologies and methodologies.

Overall, studies show that Spanish universities' practices about sustainability and energy efficiency in different areas, such as in the fields of strategic planning, practices, curricula, and reporting, show a low rate of progress, which seems like clear indication of there being several obstacles to overcome [29,30]. In fact, there are few analyses of the extent to which Spanish universities have implemented the sustainability approach in their functions. Highlights include the case of Universitat Politecnica de Catalunya, which has carried out actions to reduce energy and raw material use and drive waste out of the system [13] and the University of Santiago de Compostela's ecological footprint

since 2005 [26]. The incorporation of sustainability in curriculum in the University of Valencia [31] is also worth mentioning.

## 3. Methodology

*Documental data.* To understand the Energy Efficiency Program carried out at UAH in Madrid, Spain and its main results, Section 4 presents the UAH's Environmental Quality Program and its main strategic lines, in addition to the energy saving and conservation initiatives developed since 2006.

*Quantitative analysis.* The analyses of the buildings' energy consumption performance indicators are presented in order to offer a characterization of the results obtained in terms of energy savings. These indicators are the energy consumption in terms of electricity (Kwh) and natural gas consumption (Kwh).

In addition, a set of Energy Efficiency Indexes (*EEI*) were estimated to explore the evolution of electricity and natural gas consumption per user (Kwh/user) and per area (Kwh/m$^2$) according to Equations (1) and (2). The data used referred to period between 2003 and 2017.

$$EEI = \frac{Total\ energy\ used\ (Kwh)}{Total\ users} \tag{1}$$

$$EEI = \frac{Total\ energy\ used\ (Kwh)}{Gross\ floor\ area\ (m^2)} \tag{2}$$

Additionally, the UAH's contribution to climate change, from estimation of its $CO_2$ emissions, is discussed. The available data include the estimations until to 2014 and currently the University is working to update the results until 2018.

Last, the evolution of the international position of UAH in Green Metrics World University [32] and Coolmyplanet Rankings [33] on environmentally sustainable practices, is analyzed.

Because the methodology approach is descriptive, the results may lack generalizability. However, the UAH's case analyzed has potential, being referent of HEIs policies of energy savings at national and international level.

*Case description.* In addition to teaching, advancing knowledge, and so forth, the UAH social commitment entails becoming a benchmark for the reaching of a sustainability model, not only in the Spanish university but also across all social, technological, and economic sectors. Its many functions include fostering habits of environmental respect and decreasing the environmental impact of human activities.

With this goal in mind, in 2003, the UAH issued its "Environmental Policy Statement" (UAH-EPS) with a view to making environmental issues part of its planning, execution, and assessment strategies. This environmental policy comprises several strategic lines:

- To prevent, reduce, and eliminate negative environmental impacts that may result from the university's activities.
- To rationalize consumption and promote increasing efficiency in the use of material and energy resources.
- To promote waste prevention and appraisal (recycling, recovery, and re-use).
- To inform, train, and make the university community aware by promoting active participation in environmental management and in enhancing the quality of the university environment.
- To monitor continuously the environmental repercussions of university activity and assess achievement of the established aims and goals.
- To maintain channels of dialog and cooperation with public and private bodies engaged in environmental issues.
- To tailor the environmental policy to the new requirements proposed by university associations at home and abroad, always with a view to continuous improvement.

- To promote in the territory of influence a policy of Environmental Excellence in Development by acting as a catalyst and assessor of such a policy in collaboration with other public and private bodies.

Because of this institutional commitment, the UAH has implemented an "Environmental Quality Program" (UAH-EQP) with the aim of identifying and controlling factors affecting the environment so as to affect continuous improvement in the University's environmental management. Under this program, several actions have been taken in various fields. These include energy saving and efficiency, economizing on water, waste management, sustainable mobility, and environmental enhancement.

The UAH-EQP also aims to promote the economic and human development of territory that is compatible with an optimum environmental quality. This social commitment is reflected in the different actions designed to make clear the environmental engagement of the organization, being aware of the environmental impact generated by its activity and, especially, its energy consumption, which is why it has been working for years to improve this aspect and reduce energy consumption.

The energy and thermal consumption of the university are distributed in an equitable way. The electric consumption is mainly due to illumination both inside and outside the buildings (in the External Scientific-Technological Campus in Figure 1 in addition to buildings and facilities also serves the outdoor lighting); office equipment and investigation; air conditioning, especially refrigeration (not available in classrooms); sanitary hot water (ACS); and other energy consuming equipment, such as kitchen appliances and cafeteria. Most of the thermal resources (natural gas) are used for heating systems. The UAH's has three campuses (the historical campus, the scientific-technological external campus and the Guadalajara's campus). The major energy efficiency interventions have been addressed in the UAH's Scientific–Technological External Campus. In this campus, the University can undertake different measures (road lighting, transformation centers, buildings, etc.). The other two are integrated within the city so that efficiency actions can only be undertaken within the buildings. However, the results obtained refer to all three campuses. Hence, the UAH's energy profile has to take into account the diversity in the typology of buildings (historical and modern), the heterogeneity in terms of teaching and research activities and the environment surrounding them (urban core and campus with its own urban infrastructure).

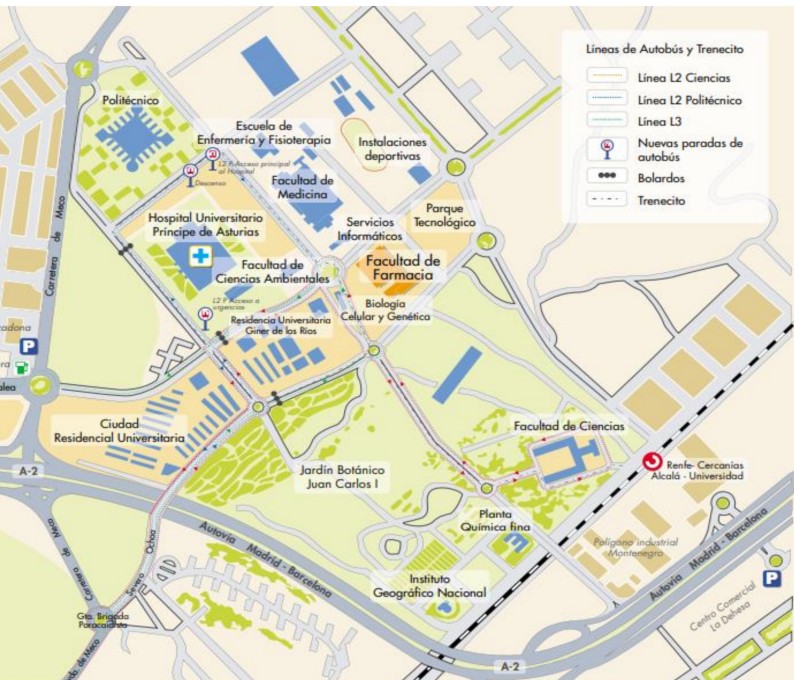

**Figure 1.** University of Alcalá's (UAH) Scientific–Technological External Campus. © The University of Alcalá.

## 4. Results

*4.1. The Trend in Energy Consumption Saving of UAH*

Since 2005 to now, the University has managed to stabilize electricity consumption, and it has registered a downturn from 2006, due chiefly to energy-saving and efficiency initiatives (see Figure 2).

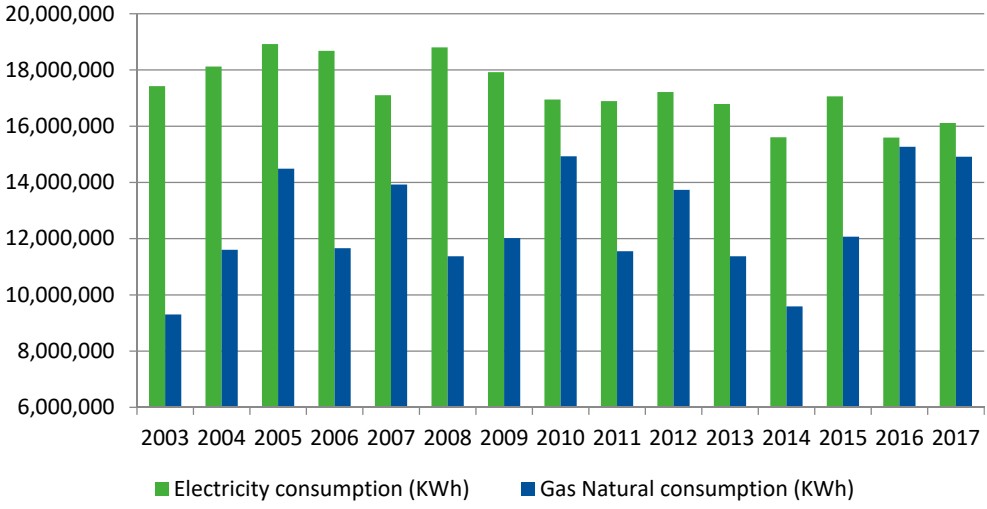

**Figure 2.** UAH's evolution of electricity and natural gas consumption. Source: Own elaboration.

The following is a synthesis of the actions taken regarding energy saving and efficiency:

1. Initiatives to raise awareness of energy saving among the university community, led chiefly by our EcoCampus office (See Ecocampus UAH. Office of Participation, Analysis and Enviromental Initiatives) [34].
2. The introduction of environmental criteria in public tender procedures (green contracting) and, particularly, when adjudicating contracts, encouragement and positive assessment of tenders made by companies that have an energy efficient management system certified by some official body.
3. Energy audits of all buildings have been performed and efficient rehabilitation work undertaken, and installations such as boilers have been replaced with others running on natural gas.
4. In addition, initiatives have been implemented in lighting systems (including sectoring of lighting in all buildings, exploitation of natural light, reduction of lighting in lifts, lighting management systems, and presence detectors in zones used occasionally) and in information systems (reduction of hidden energy consumption or stand by effect).
5. Radiators have been fitted with thermostatic valves, and solar shield panels have been installed, in addition to air curtains over either main or frequently transited entrances—all with a view to making a relevant cut in energy consumption and raising awareness among the university community.

Moreover, important efforts have been made to increase our renewable energy pool by means of thermal solar technology for ACS, the use of plant waste as biomass, the use of alternative energies for pumping, and so forth. The UAH also has an energy generating facility (Trigeneration) in its Engineering School, while its Chemistry Building boasts the most important geothermal installation in any public building in the Madrid Region and the largest of its kind in any European university (Figure 3).

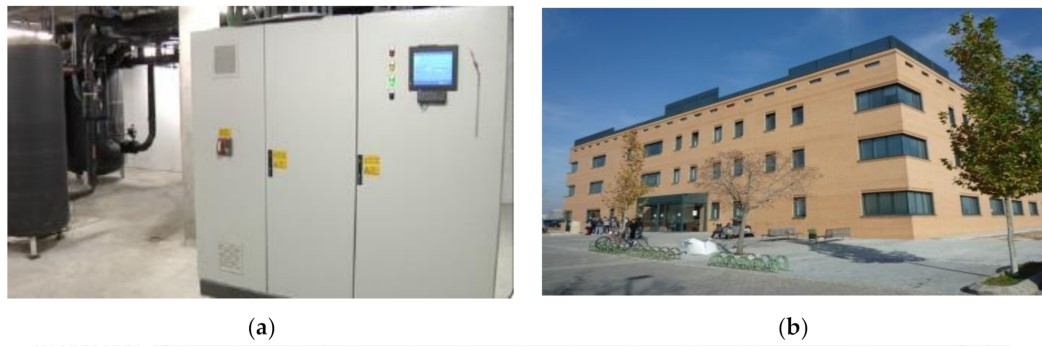

**Figure 3.** (**a**) Trigeneration Plant UAH's Polytechnic School; (**b**) UAH's Chemistry Faculty, a building of high efficiency due to the geothermal installation. © The University of Alcalá.

In recent years, however, efforts seem to have run out of steam, and there is a certain substitution effect between the two basic energy sources. There are several reasons for this. The first is the commissioning of energy-intensive facilities, such as the Learning and Research Resource Centre, an 11,000 m$^2$ heritage building, open 24 h a day, 7 days a week. Secondly, the regulatory barriers to operate the Trigeneration Plant and its technological obsolescence.

Both demand investment policies that are conditioned by budgetary restrictions or, in contrast, by new public–private partnership actions, which allow investment in efficiency by paying with the savings generated in future years.

*4.2. Main Indicators of Energy Consumption Saving of UAH in Terms of Consumption per User and Consumption per Area*

All these actions have allowed the University's energy consumption per user and per area first to fall and then to be maintained, thereby contributing to meeting the objectives of the Spanish Government's Action Plan for Energy Saving and Efficiency (2011–2020) but with a worrying change of trend in recent years (see Figures 4 and 5).

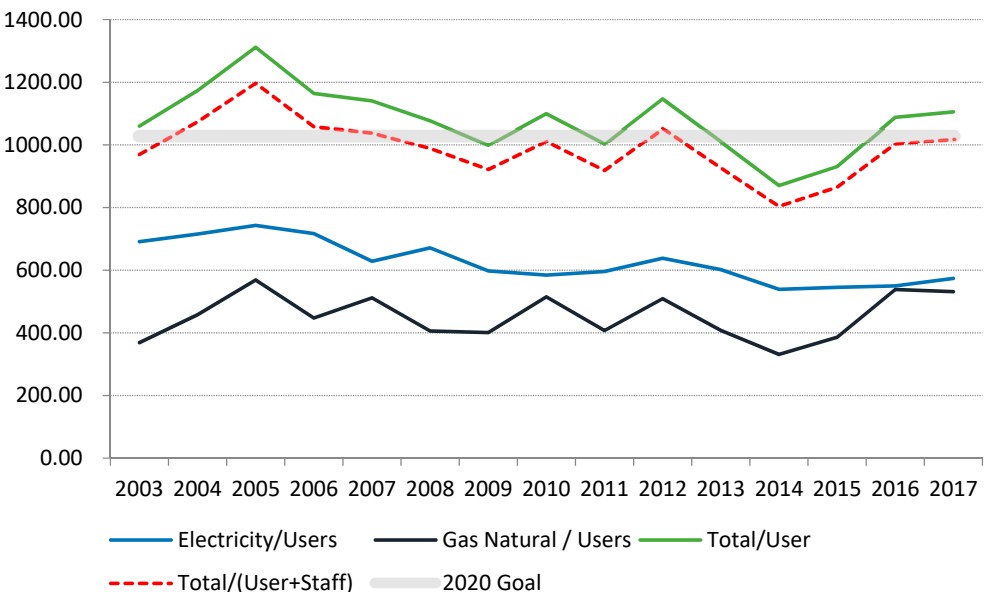

**Figure 4.** Energy consumption per user (2003–2017) and goals for 2020. Source: Own elaboration.

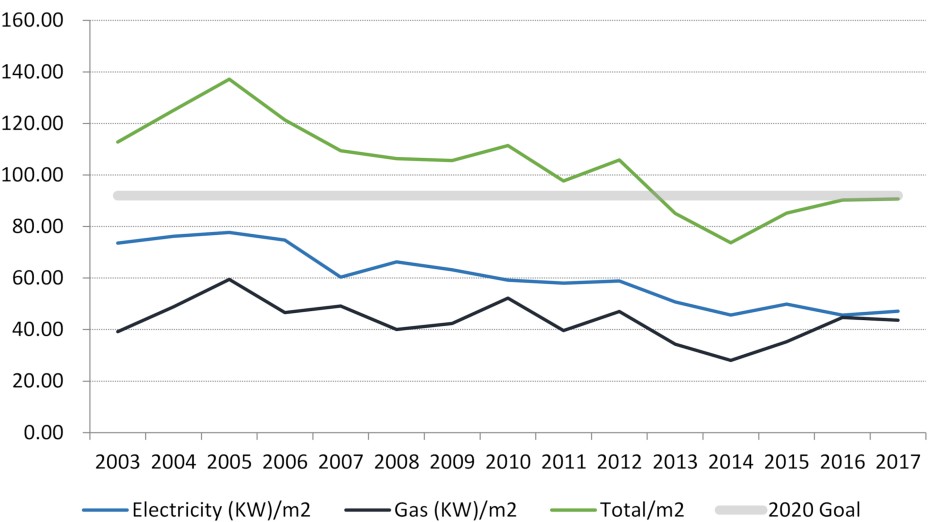

**Figure 5.** Energy consumption per area (2003–2017) and goals for 2020. Source: Own elaboration.

The number of students has fluctuated since 2010 showing a certain downward trend, with degrees now lasting 4 years instead of 5. The new energy-intensive services (24X7) increase their consumption more than proportionally to the square meters they occupy, which leads to an increase in consumption per square meter (Figure 6).

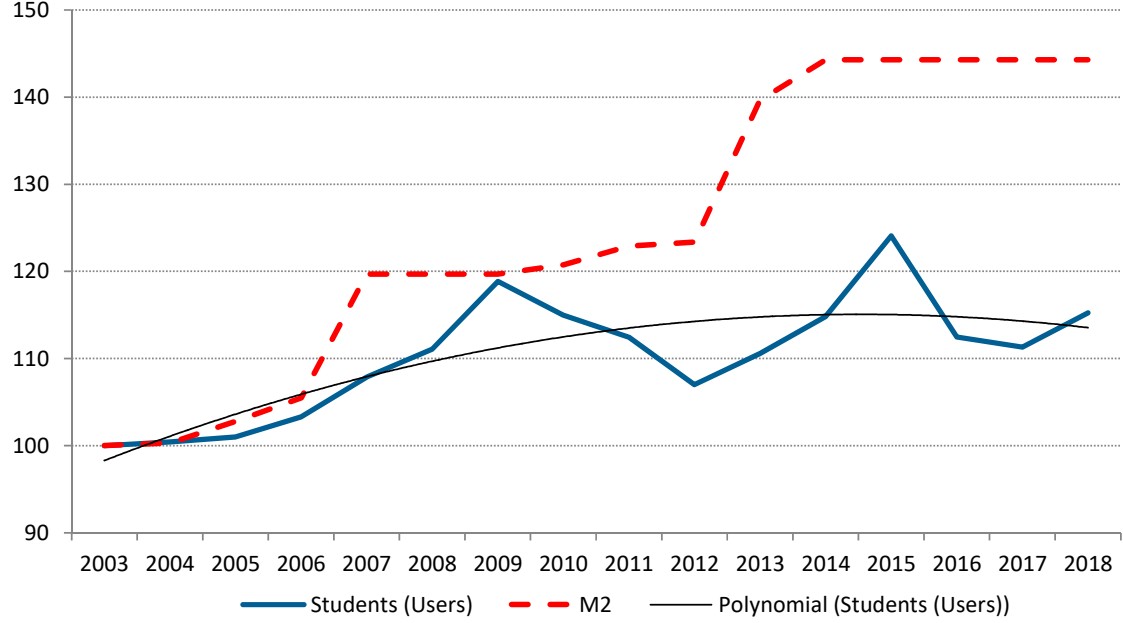

**Figure 6.** UAH's Students (clients) and Square Meters (2003=1000). Source: Own elaboration.

The UAH also boasts Spain's first solar-powered vehicle recharging point in its "Rey Juan Carlos" Botanical Garden. Two other recharging points have been opened recently at other sites (in the Pharmacy Faculty and in Malaga School in the City Campus). The Botanical Garden is, itself, a further example of sustainability, a 260,000 m$^2$ green space with more than 120,000 plant species that has become a first-class teaching and experimental resource for students and the general public alike (Figure 7).

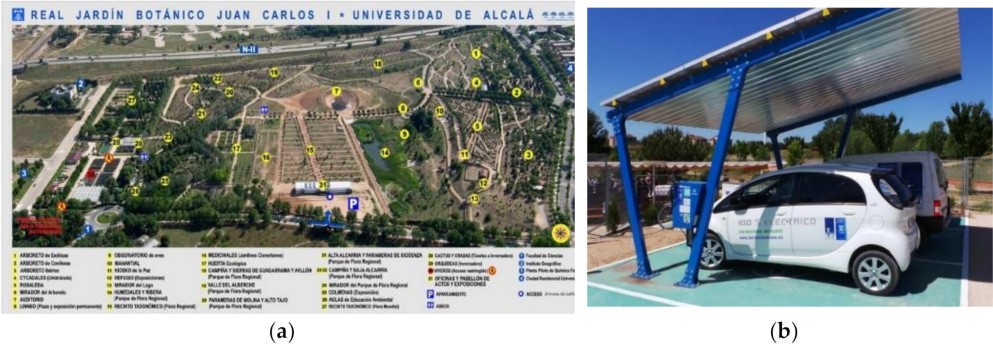

**Figure 7.** (**a**) UAH's Botanical Garden; (**b**) UAH's Solar-Powered Vehicle Recharging Points. © The University of Alcalá.

### 4.3. The Evolution of UAH's Footprint

All the UAH's initiatives carried out in the last 17 years represent a significant contribution toward building a more sustainable society, both locally and globally. In terms of sustainability, the UAH has implemented actions and measures that aim at preventing, reducing, and eliminating any negative environmental impact deriving from the university activities. However, it wants to go one step further. As a public institution devoted to education, it wants to have a transforming effect on policy and to contribute, albeit modestly, to solving global problems, including, of course, global warming.

To estimate its emissions, the UAH followed the guidelines of ISO 14064-1: 2006 and of the World Resources Institute's GHG Protocol and took into account these sources of emissions [35,36]:

- Source 1. Direct GHG emissions: Associated with combustion of natural gas in boilers, with use of refrigerant gases, and with combustion of vehicle fuel.
- Source 2. Indirect GHG emissions: Associated with electricity consumption and use of direct thermal energy.
- Source 3. Other indirect GHG emissions: Associated with travelling in buses paid for by the UAH staff, but also air and railroad travelling by UAH staff, and with fuel consumption of maintenance vehicles belonging to UAH subcontractors.

In 2014, 59.14% of the UAH's GHG net emissions were direct emissions from source 1 (2193.67 TCO2 tons of source 1), 16.37% were indirect emissions due to the consumption of electrical and thermal energy (source 2) (607.89 TCO2), and the remaining 24.49% were emissions from source 3. Emissions deriving from electricity consumption are undoubtedly the most significant (55.4% of TCO2 tons), followed by gas consumption (24.3%). That is why efficient energy policies are so important, since they reduce quite considerably the direct and indirect emissions caused by university activity (see Figure 8) (During 2020, work is underway to update the data of 2018. The restrictions resulting from the pandemic have delayed these tasks and no data are available).

In 2010, due to energy efficiency policies, the UAH was already able to consider itself a low-carbon organization, meeting even then the objectives fixed for reduction by 2040. Currently, the UAH is continuing to work on updating its carbon footprint. However, its commitment to directly reducing the emissions due to electricity usage (100 percent of renewable sources) or the compensation policies for emissions, as in the case of gas (all gas consumed is compensated) has enabled UAH to reach the 2050 targets already, with reductions above 80% in 2008–2011 and additional reductions of 25% in 2012–2014.

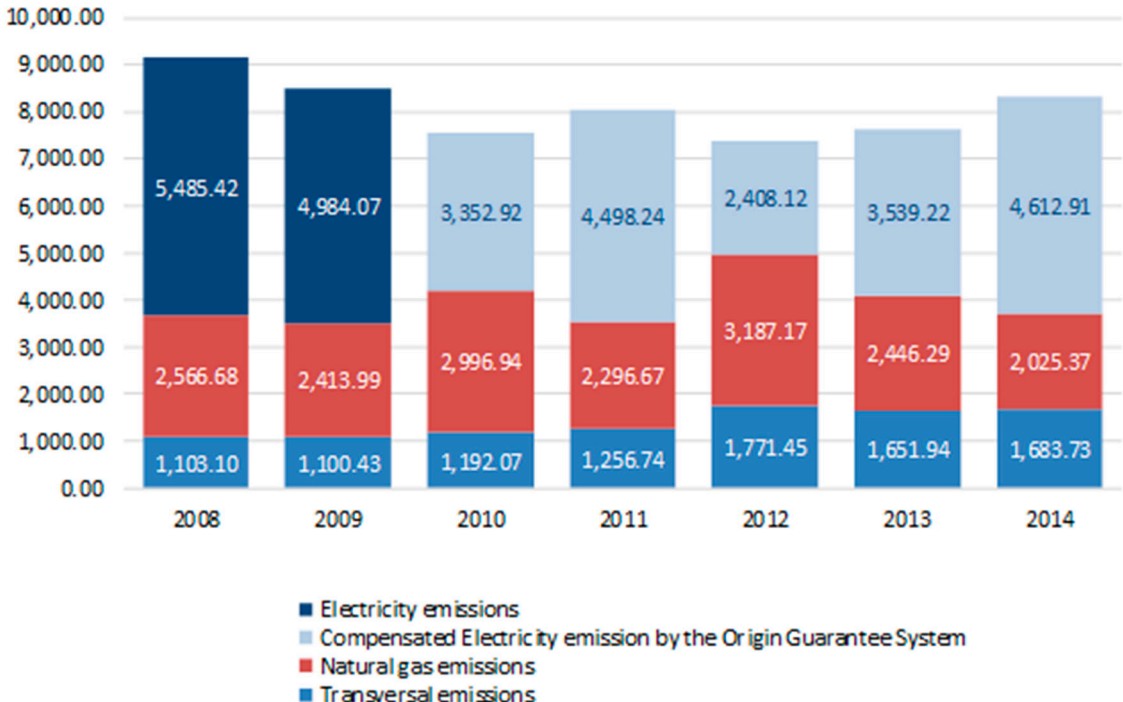

**Figure 8.** UAH's Emissions TCO2, 2008–2014. Source: Carbon footprint, UAH 2011 y 2014. Gas Natural-Fenosa—UAH [35,36].

These results encourage UAH to work even harder to improve efficiency and sustainability and to act as an example for the students, who are the generation that will have to build a more socially, economically, and environmentally sustainable future. International distinctions, such as being rated the third most sustainable university in the world by the non-profit, San Francisco-based organization, Coolmyplanet Universities Ranking [33], or its consistently high position—always among the top 40—in the University of Indonesia's Green Metric World University Rankings [32], stimulate UAH to make further improvements in the same direction (see Table 1).

**Table 1.** UAH position in Green Metrics and Coolmyplanet Rankings.

| Ranking | Year | UAH Position/Total Universities | UAH Position Regarding Spanish Universities |
|---|---|---|---|
| **Green Metrics** | 2019 | 19/780 | 2nd Spanish University |
| | 2018 | 16/719 | 1st Spanish University |
| | 2017 | 16/619 | 1st Spanish University |
| | 2016 | 26/516 | 2nd Spanish University |
| | 2015 | 37/407 | 2nd Spanish University |
| | 2014 | 28/361 | 1st Spanish University |
| | 2013 | 12/301 | 1st Spanish University |
| | 2012 | 31/215 | 1st Spanish University |
| | 2011 | 31/178 | 1st Spanish University |
| **Coolmyplanet, 50 Most Environmentally Sustainable Universities in the world** | 2016 | 3/50 | 1st Spanish University |

Source: Own elaboration based on Green Metric World University Rankings [32] and Coolmyplanet Universities Ranking [33].

## 5. Discussion

The UAH's experience over 17 years of initiatives between 2003 and 2019 suggest that the energy efficiency policies should be stepped up in accordance with four basic lines of action:

- To bring to completion the policies for energy sustainability through renewable energies.
- To go further in saving and management issues on the principle that the most sustainable energy is the energy that is not consumed.
- To integrate technology and innovation into the strategic management of sustainable energy.
- To set up a model of management and ongoing improvement to facilitate the monitoring and certification of the efficiency policies.

To this end, in 2014 a cooperation contract was signed by the public sector (the University) and the private sector to execute a global, integrated project for the interior lighting of the UAH's buildings and installations and its external campus's road. The idea was to improve energy efficiency in terms of lighting, to reduce energy consumption, and/or to create energy generating systems for either sale or self-consumption, thereby helping to cut the UAH's energy bill. This effort has resulted in a twelve-year contract worth EUR 7 million of private funding to be financed essentially through savings achieved in energy consumption. When all initiatives are 100 percent operative, there will be an annual saving of KWH 3.9, amounting to a saving of more than EUR 200,000 in direct costs to the University over the ten years that the project lasts. The project rests on four basic pillars (Figure 9) [37].

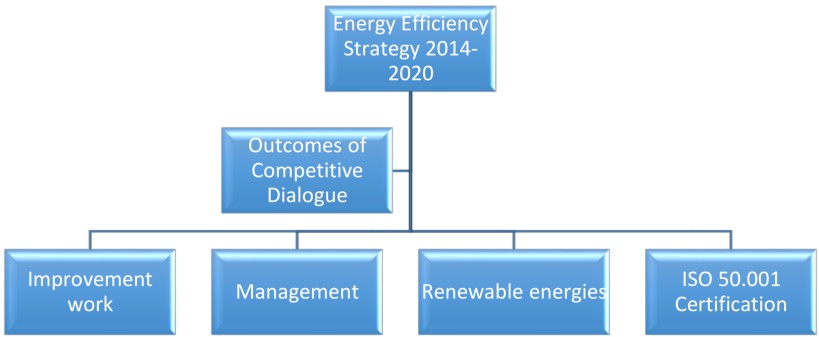

**Figure 9.** UAH's strategy regarding energy efficiency. Source: Own elaboration.

Pillar 1. Improvements and renovation work.

A series of initiatives has been envisaged on three fronts:

1. The replacement and/or renewal of both indoor and outdoor lighting installations on the External Campus. This has meant replacing more than 70,000 light fittings; it was completed in 2016.
2. The installation of building automation control and management devices (detectors, pushbuttons, sensors, light regulators, and so on) to enhance performance and efficiency while ensuring permanent user comfort.
3. Facility improvements and renovation work.

Pillar 2. Centralized energy management.

Initiatives are under way to:

1. Implement centralized energy management: quality control, uses, licenses, certifications, and regulations.
2. Develop a centralized remote Management Platform for all our facilities' building automation devices and lighting.
3. Implement a full guarantee maintenance system, with the aim of achieving operational perfection in all elements of new installations and of maintaining the performance at initial levels over time.

Pillar 3. Renewable energies.

The UAH has a significant tradition of commitment to renewable energies: geothermal energy, trigeneration, and solar panels are only some examples. Despite the scarcity of funds for investment and the absence of a stable regulatory framework, the UAH remains steadfast in its commitment to renewable energies on two fronts:

1. Encouraging direct use of renewable energies, with the installation of 4-kilowatt mini-wind farm installation (completed in 2016) that will generate electricity to be used in the Science Building, as well as a solar photovoltaic plant for the sports hall and a biomass plant at University Residential City (CRUSA) facilities. In addition, a free-cooling system is to be installed in the computing services' CPD. Furthermore, photovoltaic LED light fittings running entirely on solar power are being installed. UAH aims to promote the use of electric vehicles by increasing the number of top-up points and requiring our suppliers to use this type of vehicle.

2. Promoting consumption compensation for both electricity and gas. Since 2010 to now, UAH electrical energy has come from renewable sources, and it is certified as doing so by the National Commission of Markets and Competition. As for gas, since 2016, all emissions due to UAH consumption have been compensated by the supplier.

Pillar 4. Certification.

Introduction of an ISO 50001 EMS, to develop and implement the energy policy and manage all elements of UAH activities and all products or services that impinge on energy use (energy aspects).

The EMS, in conjunction with ISO 14001 (environment) and ISO 9001 (quality) management models, aims to achieve ongoing improvement in energy use and associated costs, the reduction of greenhouse gas (GHG) emissions, the proper use of natural resources, and the promotion of renewable energies.

The EMS is recommended for organizations wishing to show publicly the implementation of an EMS, their better use of either renewable or surplus energies, and the systematization of their processes in consonance with their energy policy. Thus, for the UAH, it is of vital importance that this ISO 50001 EMS be implemented, as it will be its energy and environmental blueprint for the years to come.

## 6. Conclusions

Energy efficiency has been one of the basic management strategies in the UAH up until 2018. This has meant that, at present, the UAH is a world leader in Energy and Climate Change, according to the 2019 edition of the Green Metrics ranking.

There are two key elements to achieving this sustained result over time: first, the actions of the UAH management team, and, second, the public–private collaboration considering different stakeholders.

The energy savings and efficiency actions include the creation of the EcoCampus office to raise awareness of energy saving among the university community, the introduction of environmental criteria in public tender procedures (green contracting), the energy audits of all buildings and the efficient rehabilitation works, the implementation of initiatives in lighting systems and in information systems, the installation of thermostatic valves in radiators, and solar shield panels, and the increase of its renewable energy pool by means of thermal solar technology for ACS, the use of plant waste as biomass, the use of alternative energies for pumping.

They have allowed the University's energy consumption first to fall and then to be maintained, thereby contributing to meeting the objectives of the Spanish Government's Action Plan for Energy Saving and Efficiency (2011–2020) but with a worrying change of trend in recent years. The UAH's has managed to stabilize electricity consumption, and it has registered a downturn by 13.8% from 2006 until 2017. The electricity consumption per user and per m2 has fallen by 20% and by 37%, respectively, in the same period. In recent years, however, the efforts developed to increase its renewable energy pool seem to have run out of steam. The functioning of Learning and Research Resource Centre, an 11,000 m$^2$ heritage building, open 24 hours a day, 7 days a week and the regulatory barriers to operate the Trigeneration Plant and its technological obsolescence are the main factors explain that negative trend. Its required investment policies that are conditioned by budgetary restrictions or by new public–private partnership actions, which allow investment in efficiency by paying with the savings generated in future years.

Regarding the UAH's footprint, the available data show that in 2010 the UAH was already in a position to consider itself a low-carbon organization, meeting even then the objectives fixed for

reduction by 2040. Currently, the UAH is continuing to work on updating its carbon footprint for the period 2015–2018. However, its commitment to directly reducing the emissions due to electricity usage (100 percent of renewable sources) or the compensation policies for emissions, as in the case of gas (all gas consumed is compensated) has enabled UAH to reach the 2050 targets already, with reductions above 80% in 2008–2011 and additional reductions of 25% in 2012–2014.

On the other hand, the public–private partnership signed in 2014 made it possible to mobilize very substantial financial resources of the order of EUR 7 million.

It allowed the remodeling of obsolete installations (lighting, computer servers) by introducing the latest available technologies and contributing to a management scheme in accordance with international certifications ISO 14,000 and ISO 50,000. However, efforts need to be redoubled and public–private collaboration maintained. Legislative changes and changes in University government teams have not contributed in recent years.

The UAH's energy saving actions findings from a thorough document analysis and the energy efficiency indicators are in the line of other similar studies as for example those referred to University of Coimbra [21], UK higher education institutions [4], Canadian post-secondary institutes [24] or in the case of a Malaysian public university [25]. In some of these studies a set of difficulties to analyze the impacts of energy efficiency intervention on HEIs are highlighted including lack of methodology, ambiguity with respect to energy consumption indicators, problems in establishing assessment boundaries, lack of clear targets for carbon reductions within the HEIs, among others.

The case analyzed of the UAH's energy efficiency has been guide by the Spanish Government's Action Plan for Energy Saving and Efficiency (2011–2020) to offer an objective analysis. However, carrying out a survey among the responsible of the energy efficiency program in the UAH could have given more understanding about the efficacy of internal programs carried out, the limitations faced, and the need to develop future actions as sectorization strategies of energy consumption and the modernization of infrastructures to introduce sensing mechanisms. The knowledge of the users' behavior is also key to undertake saving policies aimed at reducing further the consumption of all energy sources. The use of data and centralized management will allow, in the future, for advancing in this direction. For this reason, a survey among the academic community could give a more accurate comprehension about their patterns of energy-environmental behaviors and take action to raise awareness of energy efficiency in the campus.

Historically, the UAH has shown that a Sustainability Strategy not only corresponds to its basic Social Responsibility Policies, but that it is a profitable option, stabilizing consumptions and seeking, with industry, the most sustainable options in the market.

Efficiency actions in World Heritage buildings is a challenge for the future. The will of both university management and public authorities (archaeological and construction regulations) is necessary to provide these facilities with the most efficient technologies.

For example, geothermal energy installation on the scientific campus has proven to be an excellent solution for energy saving. This can be done on the historic campus but requires a shared vision from all stakeholders: University managers (General, Construction Officer, and Contracts Officer); local services and authorities that oversee the protection of Heritage.

Although the results cannot be generalized due to the limitations of the methodology used, the findings show that internal and external strategies including searching for alliances are essential elements to improving the contribution of the University to the fulfillment of the Sustainable Development Goals, not only because of its direct actions but also because of the multiplying effect that the University has as an agent of social change.

**Author Contributions:** R.G.-Y. served as general manager of UAH during 2010–2018 contributed data collection, method, data analysis as well as discussion and conclusion. M.-T.G.-R. wrote introduction, background, case description as well as discussion and conclusion. All authors have read and agreed to the published version of the manuscript.

**Funding:** This research received no external funding.

**Conflicts of Interest:** The authors declare no conflict of interest.

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
