# Peer review of "The Potential Role of Stakeholders in the Energy Efficiency of Higher Education Institutions"

_sustainability, doi:10.3390/su12218908_

Round 1

Reviewer 1 Report

Unlocking the potential of stakeholders in energy efficiency intervention in higher education institutions. The University of Alcalá experience

The paper analyzes the actions that the University of Alcalá in Madrid, Spain, about sustainability and efficiency.

Title: The proposed title is consistent with the content and topic discussed in the paper.

Abstract: The abstract is clear and concise and it presents the objective of the paper, the used methodology and the achieved results. Maybe justifying the novelty in the introduction could be a good option and insist on the used methodology, providing more detail about it.

Method: The research methods are clear and objectives and follow a logical structure.

Figures and tables: the quality and detail of the tables and the figures is adequate and clear, according to the discussed topic in the article.

Language: the used language is easily understood and errors are barely detected in it.

Structure: The followed sequence is logical and facilitates the understanding of the content of the text.

Innovation: The issue addressed in the text is quite innovative if we understand that sustainability actions are one of the main concerns of society.

In my view, some issues addressed are rather general and needs further explanations and clarifications:

  1. It is important to reduce the percentage of plagiarism because it is high, as seen in the attached report (10%).
  2. The introduction is considered as one of the main parts of the studies in which it tries to give details of the studies carried out as well as to justify the relevance of the subject and the novelty it represents for the research. Although, in the introduction of this paper, there are less relevant citations that justify the subject discussed as well as an adequate justification of the novelty that this research implies for research and for society in general.
  3. The methodology followed in this paper is exposed in one of the sections, although I consider that it would be convenient to explain the methodology followed in more detail.
  4. After reviewing the discussion and conclusions sections, I consider that they are poor in detail. They limit themselves to answering the research questions in a summarized way. It would be necessary to provide a broader discussion of the results obtained, in which the results obtained in the research are presented in a reasoned manner and the degree of novelty provided by the research carried out.

For all this, I recommend the publication of the article but with mayor modifications.

Reviewer 2 Report

Dear Authors,

Thank You for the opportunity of reading this article. My general opinion about the article is positive. 

General statements about the article

-> the article concerns the problem of the potential to save energy in Universities. This problem is very actual and desirable. The vulnerable part of the article is analyzed real case study based on the University of Alcalá.

-> article suite to Sustainability journal scope.

-> Literature background is sufficient. It's based on 36 literature positions. The majority of them are actual. References come from good quality journals.

-> Abstract is well written and includes all needed information.

-> article contribution is clearly indicated as " The paper highlights two key element in the development of HEIs' energy savings actions. First, the actions of management teams, and,  second, the public–private collaboration considering different stakeholders. In addition, this paper fulfils an identified need to study the impact of HEIs in sustainable development through initiatives in saving energy in their campus. 

-> quality of figures and tables is high. 

-> The good part of the article is the discussion part. Conclusions are clearly written.

I will recommend publishing this article after minor revision which consists of:

#1
Please use "[]" for literature citation in the text not "()"
#2
Please add doi for the articles in the literature review. Actually, it's strongly recommended to add doi in references. So please add to improve the quality of references.
#3
Please introduce the abbreviation UN - line 31

Reviewer 3 Report

The main drawback of the article are the following:

Title: The title is too long and does not capture the importance of the study and the attention of the reader.

Abstract: This section must to be rewrite. The ideas of the research could have been more effective through the use of elaborative and concise sentences. The abstract as is, does not provide a concise account of the work and conclusion of the research study. It needs to be more structured and synthesized for research clarity.

Literature review: The 36 references are appropriate, but they are not well integrated and critically analysed.
The title of the section - 2.3. Case description – does not reflect the content of the section and if it is a Case description, but it does not - the place of it seems to be in Methodology and not in a literature review section.

What is the significance of the Figure 1. UAH’s Scientific–Technological External Campus?

The place of the two Research questions is in Introduction section and not in Methodology.

The scientific methodology is not properly described.

There is lack of clear evaluation methods and results and there is lack of quantifiable description of the findings.
The discussion section needs more substance.

Good luck!

Round 2

Reviewer 1 Report

After reviewing the new contributions of the authors, I consider that it has been substantially improved.

Reviewer 3 Report

The paper was improved, but the title is not so suggestive for the content of the article.